# Patterns of *Kdr*-L995F Allele Emergence Alongside Detoxifying Enzymes Associated with Deltamethrin Resistance in *Anopheles gambiae* s.l. from North Cameroon

**DOI:** 10.3390/pathogens11020253

**Published:** 2022-02-15

**Authors:** Josiane Etang, Stanislas Elysée Mandeng, Philippe Nwane, Herman Parfait Awono-Ambene, Jude D. Bigoga, Wolfgang Eyisap Ekoko, Achille Jerome Binyang, Michael Piameu, Lili Ranaise Mbakop, Narcisse Mvondo, Raymond Tabue, Rémy Mimpfoundi, Jean Claude Toto, Immo Kleinschmidt, Tessa Bellamy Knox, Abraham Peter Mnzava, Martin James Donnelly, Etienne Fondjo

**Affiliations:** 1Laboratoire de Recherche sur le Paludisme, Institut de Recherche de Yaoundé (IRY), Organisation de Coordination pour la lutte Contre les Endémies en Afrique Centrale (OCEAC), P.O. Box 288, Yaoundé 999108, Cameroon; mandengelysee@yahoo.fr (S.E.M.); philino07@yahoo.fr (P.N.); hpaawono@yahoo.fr (H.P.A.-A.); ewolfgang388@gmail.com (W.E.E.); piameujr@yahoo.fr (M.P.); mbalira@yahoo.fr (L.R.M.); jctotofr@yahoo.fr (J.C.T.); 2Department of Biological Sciences, Faculty of Medicine and Pharmaceutical Sciences, University of Douala, P.O. Box 2701, Douala 999108, Cameroon; 3Institute for Insect Biotechnology, Justus Liebig University Gießen, 35394 Gießen, Germany; 4Laboratory of Animal Biology and Physiology, Faculty of Sciences, University of Yaoundé I, P.O. Box 337, Yaoundé 999108, Cameroon; bjeromeachille@yahoo.fr (A.J.B.); nar6mv@yahoo.fr (N.M.); remy.mimfoundi@yahoo.fr (R.M.); 5Laboratory for Vector Biology and Control, National Reference Unit for Vector Control, The Biotechnology Center, Nkolbisson-University of Yaounde I, P.O. Box 3851 Messa, Yaoundé 999108, Cameroon; judebigoga@yahoo.com (J.D.B.); tnraymon@yahoo.fr (R.T.); 6Ecole des Sciences de la Santé, Université Catholique d’Afrique Centrale, P.O. Box 1110, Yaoundé 999108, Cameroon; 7National Malaria Control Programme, Ministry of Public Health, Yaoundé 999108, Cameroon; 8MRC International Statistics and Epidemiology Group, Department of Infectious Disease Epidemiology, London School of Hygiene and Tropical Medicine, Keppel St., London WC1E 7HT, UK; immo.kleinschmidt@lshtm.ac.uk; 9Wits Research Institute for Malaria, School of Pathology, Faculty of Health Sciences, University of the Witwatersrand, Johannesburg 2141, South Africa; 10Southern African Development Community Malaria Elimination Eight Secretariat, 10 Platinum Street, Erf 490, Prosperita, Windhoek 10005, Namibia; 11World Health Organization Country Liaison Office, Port Vila 99514, Vanuatu; knoxt@who.int; 12African Leaders Malaria Alliance (ALMA), Dar es Salaam 11101, Tanzania; drabrahammnzava@gmail.com; 13Department of Vector Biology, Liverpool School of Tropical Medicine, Pembroke Place, Liverpool L3 5QA, UK; martin.donnelly@lstmed.ac.uk; 14ABT ASSOCIATES, PMI VectorLink, Yaoundé 999108, Cameroon; fondjoetienne@yahoo.fr

**Keywords:** *Anopheles gambiae* s.l., deltamethrin, *Kdr* 995F/S alleles, synergists, insecticide resistance, malaria vector control, resistance management, northern Cameroon

## Abstract

Understanding how multiple insecticide resistance mechanisms occur in malaria vectors is essential for efficient vector control. This study aimed at assessing the evolution of metabolic mechanisms and *Kdr* L995F/S resistance alleles in *Anopheles gambiae* s.l. from North Cameroon, following long-lasting insecticidal nets (LLINs) distribution in 2011. Female *An. gambiae* s.l. emerging from larvae collected in Ouro-Housso/Kanadi, Be-Centre, and Bala in 2011 and 2015, were tested for susceptibility to deltamethrin + piperonyl butoxide (PBO) or SSS-tributyl-phosphoro-thrithioate (DEF) synergists, using the World Health Organization’s standard protocol. The *Kdr* L995F/S alleles were genotyped using Hot Ligation Oligonucleotide Assay. Tested mosquitoes identified using PCR-RFLP were composed of *An. arabiensis* (68.5%), *An. coluzzii* (25.5%) and *An. gambiae* (6%) species. From 2011 to 2015, metabolic resistance increased in Ouro-Housso/Kanadi (up to 89.5% mortality to deltametnrin+synergists in 2015 versus <65% in 2011; *p* < 0.02), while it decreased in Be-Centre and Bala (>95% mortality in 2011 versus 42–94% in 2015; *p* < 0.001). Conversely, the *Kdr* L995F allelic frequencies slightly decreased in Ouro-Housso/Kanadi (from 50% to 46%, *p* > 0.9), while significantly increasing in Be-Centre and Bala (from 0–13% to 18–36%, *p* < 0.02). These data revealed two evolutionary trends of deltamethrin resistance mechanisms; non-pyrethroid vector control tools should supplement LLINs in North Cameroon.

## 1. Background

Since the discovery of chemical insecticides, there has been an increasing dependence on synthetic compounds for vector-borne diseases prevention [1], owing to their effectiveness and ease of use. Among the vector-borne diseases is malaria, an acute febrile illness caused by *Plasmodium* parasites transmitted by *Anopheles* mosquitoes [2]. Its mass prevention heavily relies on the use of pyrethroid insecticides in long-lasting insecticidal nets (LLINs) and indoor residual spraying (IRS) interventions [1,3]. During the last two decades, the rapid scaling-up of LLINs and IRS interventions has largely contributed to the decline of morbidity and mortality related to malaria [4]. In sub-Saharan Africa, malaria case incidence was reduced from 368 to 222 per 1000 population at risk between 2000 and 2019, but increased to 232 per 1000 in 2020, mainly because of disruptions to services during the COVID-19 pandemic. Malaria deaths in this region were also reduced by 36%, i.e., from 840,000 in 2000 to 534,000 in 2019, before increasing to 602,000 in 2020 [3].

However, whilst at least half of the African population is now sleeping under LLINs [5], pyrethroid resistance has become a common phenomenon in the most efficient vectors of *Plasmodium* parasites, especially those belonging to the *Anopheles gambiae* complex and the *An. funestus* group [6]. Insecticide resistance mechanisms are strongly selected following the use of chemicals in agriculture and public health interventions, leading to the reduction of the effectiveness of LLINs and IRS. To date, two types of pyrethroid resistance mechanisms are mostly recorded, being target site modifications and increased detoxification, allowing resistant mosquitoes to metabolize insecticides at a higher rate. The target site modifications result from single-base mutations in the voltage-gated sodium channel (*Vgsc*-L995S/F) gene—the target site for pyrethroids and DDT, leading to the knockdown resistance (*kdr*) L995F, L995S, and N1570Y mutations [7,8]. The L995F mutation, a leucine to phenylalanine change, has rapidly disseminated across three species of the *An. gambiae* complex, i.e., *An*. *arabiensis*, *An*. *coluzzii* and *An. gambiae* s.s. However, its distribution is not uniform either within or between these species. The reasons for this divergence remain unclear because little is known about the adaptive capacity that this mutation confers to wild mosquito populations under selection pressure in the field. On the other hand, metabolic resistance arises from increased detoxification of insecticides, by either of the three classes of metabolic enzymes, carboxylesterases (COEs), glutathione-S-transferases (GSTs), and cytochrome P450s [9]. Overexpression of several P450 monooxygenases has been associated with insecticide resistance, and those which have been validated as pyrethroid-metabolizers in vitro, and thus capable of causing resistance include CYP6M2, CYP6P3, and CYP6P4 [10,11,12]. Carboxylesterases can also play a role in pyrethroid metabolism, for example, when coupled with P450s [13,14]. Furthermore, pyrethroid resistance may also be mediated by glutathione *S*-transferases (GST) GST1-6 [15].

From previous research, multiple insecticide resistance mechanisms, including target site and metabolic mechanisms, have been documented in *Anopheles* mosquito species across many countries in Sub-Saharan Africa, including Cameroon [16,17,18,19]. However, it remains unclear whether these two major types of resistance mechanisms are similarly influenced by selective pressure in the field, or one mechanism might evolve faster and dominate the others. The resulting evolutionary trend of resistance mechanisms is of particular importance since there is evidence that the wide-scale implementation of IRS and LLINs is contributing to the selection of pyrethroid resistance in the major African malaria vectors [20,21]. Understanding how multiple insecticide resistance mechanisms arise and the evolution of associated survival advantage in mosquito populations could help to better manage vector resistance to insecticides and sustain vector control interventions.

In Cameroon, malaria prevention relies on the wide use of LLINs. The deployment of complementary measures, such as IRS with insecticides of other classes than pyrethroids as recommended by WHO [22] is planned for the near future [23]. The National Malaria Control Program (NMCP) conducted the very first nationwide free LLIN distribution campaigns with 8 million LLINs in 2011 [24]. At that time, metabolic resistance and *Kdr*-based resistance to pyrethroids were already reported among *An*. *gambiae* s.l. populations from several regions [25,26]. In the North Region, pyrethroid resistance in *An. arabiensis* was found to be essentially mediated by enzyme-based detoxification [26,27]. Subsequent studies have reported *Kdr* L995F allele (at up to 61% frequencies) in *An. coluzzii, An. gambiae* s.s. and *An. arabiensis* from this region [28]. However, how the expansion of the *Kdr* L995F allele arose in mosquito populations already displaying upregulated metabolic genes has not been investigated. In other words, the arrangement, and operational mode of the *Kdr* L995F allele on top of metabolic mechanisms in local vector populations remains poorly understood. Considering the predominance of metabolic-based resistance mechanisms in *An. arabiensis* populations from North Cameroon in 2011, we set up to investigate the changes in the patterns of the *Kdr* resistance alleles and metabolic detoxication activities and the resulting phenotypes of pyrethroid resistance, following the wide distribution of LLINs in 2011. Besides, this region is subject to extensive use of different organophosphate and pyrethroid insecticides in cotton cultivation areas. We hypothesized that such a diversified selective pressure might have different impacts on the resistance mechanisms in place.

The current study aimed at assessing the patterns of the *kdr* 995F/S resistance alleles’ emergence alongside cytochrome P450s and COEs detoxifying enzyme activities in pyrethroid-resistant *An. gambiae* s.l. from North Cameroon. This study is part of a multi-country project that assessed the impact of insecticide resistance on the effectiveness of LLINs or IRS interventions [29].

## 2. Results

### 2.1. Distribution of Mosquito Species across the Study Sites

Mosquitoes that were morphologically identified as belonging to the *An. gambiae* complex and used as control during deltamethrin susceptibility tests were identified down to species using molecular identification PCR-RFLP techniques. These include 200 mosquito specimens of the Kisumu laboratory reference strain and 352 field mosquitoes. Among these samples, 223 were randomly selected in 2011 (N = 100 for the Kisumu strain and 123 from the field) and 329 in 2015 (N = 100 for the Kisumu strain and 229 from the field).

The samples of the Kisumu reference were solely composed of the *An. gambiae* species; while three species of the *An. gambiae* complex, namely *An. arabiensis*, *An. coluzzii* and *An. gambiae* were recorded in field mosquito samples (Figure 1). In general, *An. arabiensis* was the dominant species, representing 73.2% and 52.4% of the samples collected in 2011 and 2015, respectively. This dominance was also reflected in each study site, except in Ouro-Housso/Kanadi where *An. coluzzii* became the dominant species in 2015 (58%).

The distribution of the three species varied from site to site and from one period to another. In Be-Centre and Bala located in sub-urban (Pitoa HD) and rural (Mayo-Oulo HD) areas, respectively, *An. gambiae* which was not recorded in 2011 appeared in 2015, at 17% and 6% frequencies, respectively. Furthermore, *An. coluzzii* was recorded in Bala only in 2015 (11% frequency) though absent in 2011, highlighting the progressive increase of species diversity in rural and semi-urban areas. However, in Ouro-Housso/Kanadi located in an urban area (Garoua HD), the three species were recorded both in 2011 and in 2015, but with a decreasing frequency for *An. arabiensis* and increasing frequencies for *An. coluzzii* and *An. gambiae*.

### 2.2. Deltamethrin Resistance Profiles

A total of eight deltamethrin susceptibility tests were carried out, four in 2011 and four in 2015 i.e., one assay with each mosquito strain per year. The knockdown times (kdt) during exposure to deltamethrin and the mortality rates 24 h post-exposure are provided in Table 1 and Figure 2, respectively.

The kdt for 50% mosquitoes (kdt_50_) of the Kisumu susceptible strain (N = 172) were below 10 min; the corresponding kdt_95_ were below 20 min and the 24-h mortality rates were 100%.

However, field mosquito populations (N = 532) displayed variable trends of kdt and mortality rates. In Ouro-Housso/Kanadi, the kdt_50_ was over 50 min in 2011 and in 2015, corresponding to >5-fold those of the Kisumu susceptible strain. In Be-Centre, the kdt_50_ increased from 23.5 min in 2011 to 37 min in 2015, suggesting a decrease of deltamethrin kd effect. In Bala, it conversely decreased from 41 min in 2011 to 28 min in 2015, suggesting an increase of deltamethrin kd effect. The non-overlapping kdt_50_ confidence intervals in Be-Centre and Bala between 2011 and 2015, suggest that the observed variations of knockdown effects were more than mere sampling variations. For all assays with field-collected mosquitoes, kdt_95_ were >60 min in 2011 and in 2015. Overall mortality rates were significantly lower in Ouro-Housso/Kanadi (40.1%) compared to Be-Centre (86.4%) and Bala (83.4%) (*p* < 0.001), suggesting high resistance frequencies in urban areas (Ouro-Housso/Kanadi) compared to sub-urban (Be-Centre) and rural areas (Bala). More specifically, the resistance status was regularly confirmed in *An. gambiae* s.l. samples from Ouro-Housso/Kanadi (<50% mortality) and Bala (82–85% mortality). However, in Be-Centre, resistance suspected in 2011 (90% mortality) was confirmed in 2015 (83% mortality).

### 2.3. Metabolic Resistance Indicators: Effects of Synergists on Deltamethrin Resistance

The piperonyl butoxide (PBO) and SSS-tributyl-phosphoro-thrithioate (DEF) synergists were used for the detection of P450 and COE detoxifying enzymes’ involvement in deltamethrin resistance. A total of 14 susceptibility tests were carried out with deltamethrin-synergist combinations, whereby mosquitoes (N = 1807) were exposed to PBO (N = 619) or DEF (N = 484) synergists prior to deltamethrin susceptibility tests. Four assays were conducted with mosquitoes of the Kisumu reference susceptible strain and 10 assays with wild *An. gambiae* s.l. samples. In Ouro-Housso/Kanadi and Be-Centre, the 3 test configurations (deltamethrin, PBO+deltamethrin, and DEF+deltamethrin) were successfully carried out; while in Bala, tests with DEF+deltamethrin were not carried out because of the scarcity of mosquito larvae in breeding sites.

Different profiles of synergistic effects of PBO and DEF were observed in the kdt and mortality rates; data are shown in Table 2 and Figure 3, respectively. The suppression of kdt (kdts_50_) as a result of pre-exposure of mosquitoes to synergists was not systematically associated with an increase of their mortality rates (Mrt) to deltamethrin and vice versa.

#### 2.3.1. Effects of Synergists on Mosquito Knockdown to Deltamethrin

With the Kisumu laboratory strain of *An. gambiae*, no significant difference of kdt was observed between the tests with deltamethrin alone and those with deltamethrin-PBO or -DEF combinations. The rates of kdts_50_ induced by synergists were less than 10% (Table 2).

With *An. gambiae* s.l. samples from Ouro-Housso/Kanadi, the Kdt_50_ of the synergist+deltamethrin combination assays were mostly reduced compared with deltamethrin alone, except with DEF in 2011 (59 min); however, they remained above 40 min. Relative reductions in knockdown times, kdts_50_, varied between 30% and 56%, except with DEF in 2011 (kdts_50_ < 0). The kdt_95_ were all greater than 60 min, suggesting no detectable effect of synergists on this variable.

With *An. gambiae* s. l. samples from Be-Centre, pre-exposure to PBO and DEF synergists generally resulted in a significant reduction of kdt_50_ from 23–37 min to less than 15 min with corresponding kdts_50_ rates between 54% and 71%. The kdt_95_ were also decreased from >60 min to 25–42 min only in 2011, but not in 2015. More specifically, there was no synergistic activity of DEF on deltamethrin kdt in 2015 (kdts_50_ rate < 0%), and both kdt_50_ and kdt_95_ remained >60 min.

With *An. gambiae* s. l. samples from Bala, the use of PBO synergist did not induce any significant reduction of kdt_50,_ either, in 2011 (38 min) or in 2015 (26 min); the kdt_95_ remained >50 min and the kdts_50_ relative reductions were less than 10%.

#### 2.3.2. Effects of Synergists on Mosquito Mortality to Deltamethrin

The mortality rates of the Kisumu reference strain of *An. gambiae* to deltamethrin with or without synergists were always 100% (Figure 2).

With field mosquito samples, pre-exposure to PBO synergist significantly increased their mortality to deltamethrin, both in 2011 and in 2015 (6.8 < X^2^ < 49.5; df = 1; *p* < 0.02), except in a few cases. When comparing mortality rates between 2011 and 2015, two trends were observed in the synergistic effects of PBO and DEF. In the Ouro-Housso/Kanadi urban area, there was an increase in PBO and DEF synergistic effects over the time. The mortality induced by deltamethrin when combined with either of the synergists were significantly increased in 2015 (73% for PBO-deltamethrin and 89.5% DEF-deltamethrin) compared with those recorded in 2011 (64% for PBO-deltamethrin and 45% for DEF-deltamethrin) (X^2^ = 6.0; df = 1; *p* < 0.02). Conversely, the effects of synergists obviously declined over the years in the Be-Centre sub-urban and Bala rural areas, with mortality rates lower in 2015 (42–94%) than in 2011 (>95%) (X^2^ = 61.7; df = 1; *p* < 0.001).

In terms of resistance status (Table 2), *An. gambiae* s.l. samples from Ouro-Housso/Kanadi maintained their “confirmed resistance” status, regardless of pre-exposure to synergists either in 2011 or in 2015. Similarly, when mosquito samples from Be-Centre were pre-exposed to DEF, the resistance status remained “suspected resistance” and “confirmed resistance” in 2011 and 2015, respectively. However, the reversion of resistance status (from “suspected resistance” to “susceptible”) following pre-exposure to synergists observed in Be-Centre in 2011, was no longer effective in 2015. Furthermore, a regression of PBO synergistic effect on the reversion of resistance status was observed in Bala, from complete reversion in 2011 (“confirmed resistance” to “susceptible”), to partial reversion in 2015 (i.e., “confirmed resistance” to “suspected resistance”).

### 2.4. Genotypic and Allelic Frequencies at the Kdr 995 Locus

The L995F *Kdr* mutation was found in the three species of the *An. gambiae* complex, namely *An. arabiensis*, *An. coluzzii* and *An. gambiae*, but at variable genotypic and allelic frequencies (Figure 3). However, the *kdr* L995S mutation was not recorded in any of analyzed specimens.

Both homozygous *Kdr* 995L/L (susceptible) and *Kdr* 995F/F (resistant), as well as the heterozygous *Kdr* 995L/F genotypes, were found in the three study sites. In Ouro-Housso/Kanadi, the genotypic frequencies of homozygous resistant decreased from 25–40% in 2011 to 9–20% in 2015; this decrease was observed in each of the three sibling species. Conversely, there was an increase of heterozygote frequencies from 5–63% in 2011 to 22–79% in 2015. In Be-Centre, the homozygous resistant genotype observed mainly in *An. coluzzii* in 2011 at 42.8% frequency was no longer recorded in 2015, neither in this species nor in the two other sibling species. Instead, the frequencies of heterozygotes in the three species increased from 0–29% in 2011 to 64–87% in 2015. Simultaneously, there was a reduction in the frequency of susceptible homozygotes. Furthermore, the frequency of heterozygotes in mosquito samples from Bala increased from 0% in 2011 to between 24–89% in 2015. In this study site, homozygous resistant mosquitoes were only observed in the *An. gambiae* species in 2015, with up to 20% genotypic frequencies.

The *Kdr* L995F allelic frequencies in analyzed samples of *An. gambiae* s.l. are summarized in Table 3. We displayed the *Kdr* L995F frequencies for each of the three identified species (*An. arabiensis*, *An. coluzzii,* and *An. gambiae*), as well as the overall frequencies at the level of *An. gambiae* s.l. complex (i.e., pooled sampled including the three species).

Either in *An. arabiensis*, *An. coluzzii* or in *An. gambiae*, the allelic frequencies of *kdr* L995F decreased between 2011 and 2015 in Ouro-Housso/Kanadi, although the difference was not significant (*p* ≥ 0.35). Conversely, they increased from 3.33% to 35.71% and from 0.00% to 13.32% in *An. arabiensis* from Be-Centre and Bala, respectively (0.003 ≤ *p* ≤ 0.04). Due to the scarcity of *An. coluzzii* and/or *An. gambiae* in mosquito samples from Be-Centre and Bala in 2011, it was not possible to assess the changes in the *Kdr* 995F allelic frequencies in 2015. When pooling the samples of the three species, *An. gambiae* s.l. samples from Ouro-Housso/Kanadi displayed a non-significant decrease of *kdr* L995F allelic frequency from 50% in 2011 to 45.7% in 2015 (*p* = 0.88). By contrast, the frequency of this resistant allele significantly increased between 2011 and 2015 in the two other study sites, i.e., from 13.5 to 35.8%, and from 0 to 18.1% in Be-Centre and Bala, respectively (0.01 ≤ *p* ≤ 0.02).

### 2.5. Interactions between Metabolic and kdr 995F Resistance Mechanisms

To further explore the interplay between insecticide resistance mechanisms in the studied *An. gambiae* s.l. populations, we compiled and considered the data on the kdts_50_ and Mrt. to deltamethrin with or without synergists and the frequencies of the *Kdr* L995F resistance allele.

We found two patterns of multiple resistance mechanisms development, involving mainly P450s and the *Kdr* L995F mutation, associated to COEs to a lesser extent (Table 4). In Ouro-Housso/Kanadi and Bala, the “confirmed resistance” status remained unchanged between 2011 and 2015; however, the resistance mechanisms that evolved over the years were different between the two study sites. In the presence of the *Kdr* L995F mutation as the primary resistance mechanism in Ouro-Housso/Kanadi, an increase of kdts_50_ and Mrt. to deltamethrin following pre-exposure of mosquitoes to PBO and DEF in 2015 as compared with 2011, suggest an upsurge of detoxification by P450s and COEs. In contrast, in the presence of P450s as the primary resistance mechanism in *An. gambiae* s.l. from Bala, the mechanism of resistance that emerged was the *Kdr* L995F allele.

More interestingly, with the *An. gambiae* s.l. samples from Be-Centre, where the resistance status changed from “suspected” to “confirmed” resistance between 2011 and 2015, the effects of synergists in terms of kdts_50_ (with DEF) and restoration of Mrt (with PBO) were rather regressive, in parallel with an increase in the frequency of the *Kdr* L995F allele.

## 3. Discussion

In sub-Saharan Africa, three malaria vector species belonging to the *An. gambiae* complex, namely *An. gambiae*, *An. coluzzii* and *An. arabiensis* are mostly exhibiting multiple insecticide resistance mechanisms. However, little is known about the circumstances and speed of development of multiple resistance mechanisms in the field. When there is selection pressure, there may be interactions between the different mechanisms in place. Here, we report an apparent regression of metabolic resistance in favor of the *Kdr* L995F mutation in *An. gambiae* s.l. from semi-urban and rural areas in North Cameroon between 2011 and 2015. Conversely, there is an increase of metabolic detoxification over the *Kdr* L995F allele in urban areas. These findings are considered to help decision-makers when having to prioritize complementary malaria vector control interventions in Cameroon. One limitation of this study is that, because of the low numbers of mosquito larvae collected from breeding sites, the sample size of *An. gambiae* s.l. used as control during susceptibility tests and subsequently for species identification and *Kdr* L995F/S genotyping was not large. This might have affected the data on species distribution. For the same reason, we were not able to conduct tests with DEF+deltamethrin in Bala in either of the two study years.

Nevertheless, the presence of *An. gambiae*, *An. coluzzii* and *An. arabiensis* species in Ouro-Housso/Kanadi, Be-Centre, and Bala is in agreement with previous studies [31,32,33]. Moreover, we noted an increase in species diversity over time and variable distribution patterns of the three species across the study sites. In 2015, *An. coluzzii* became a major malaria vector species in Ouro-Housso/Kanadi and Be-Centre. Additionally, *An. gambiae* which was not recorded in Be-Centre and Bala in 2011 appeared there in 2015. It is possible that environmental factors, such as the types of mosquito breeding water bodies, their surroundings, and the origin of water, in combination with the ability of mosquito species to develop insecticide resistance, lead to the emergence and predominance of one taxonomic group rather than the others. *An. coluzzii* and *An. gambiae* are highly adaptive malaria vectors colonizing both tropical and equatorial areas [33], while *An. arabiensis* is known to colonize tropical areas from West to East Africa. The low proportion of *An. gambiae s.s.* in the three study sites could also be explained by the fact that larvae were collected in September and November, at the end of the rainy season. This species prefers small, temporal, and clean water bodies formed by the accumulation of rainwater, whereas *An. coluzzii* is usually found in permanent and semi-permanent breeding sites, such as those resulting from human activities in urban areas [34,35]. The urbanization gradient of the surveyed areas could also explain the increase of *An. coluzzii* in samples from Be-Centre and Bala in 2015. The ability of *An. coluzzii* to adapt in polluted breeding sites with a predominance of organic matters in the urban agglomerations of Cameroon has been highlighted by Tene Fossog et al. [36].

Data on the evolution of deltamethrin resistance between 2011 and 2015 highlights a change in the resistance status from “suspected” to “confirmed” resistance in Be-Centre; while in Ouro-Housso/Kanadi and Bala, the resistance observed in 2011 was reconfirmed in 2015, with declining mortality rates. The rapid increase of insecticide resistance in *An*. *arabiensis*, *An. gambiae s.s.* and *An. coluzzii* populations from northern Cameroon have largely been demonstrated in previous studies [28,37,38,39]. Therefore, our analysis focused on the evolution and interaction between the underlying mechanisms, with particular attention to the *Kdr* L995F mutation and the reversion of resistance by PBO and DEF, as indicators of metabolic resistance conferred by P450s oxidases and COEs. The incomplete recovery of deltamethrin susceptibility after pre-exposure of mosquitoes to synergist PBO and DEF that was seen in most of the assays suggests a partial involvement of P450 and COE genes in the process of deltamethrin resistance. We found that, in addition to the reversion of mortality rates, the synergists also induced the suppression of kdt. The kdts_50_ appeared to be a reliably sensitive indicator of metabolic resistance; it has revealed the increase in the activity of P450s and COEs in Ouro-Housso/Kanadi between 2011 and 2015, as well as the decrease of COE activity in Be-Centre. However, the relationship between the kdts_50_ induced by synergists and the *Kdr* L995F resistance allele which induces an increase in kdt remains to be explored. Furthermore, *An. gambiae* s.l. mortality rate to deltamethrin + DEF in Be-Centre in 2015 was significantly lower than that recorded with deltamethrin alone. In essence, we would expect the deltamethrin + DEF mortality rate to be comparable to that of deltamethrin without DEF, in case COEs have no impact on resistance status. However, this was not the case; DEF rather acted as a deltamethrin resistance stimulus. Similar results have been recorded in previous studies [16], but the reasons for these effects are not clear. Further investigations are needed to better understand the interactions between insecticides and synergists. On top of metabolic-based deltamethrin resistance in Be-Centre and Bala, there were significant variations in the frequencies of the *Kdr* L995F allele among the three species of the *An. gambiae* complex. However, the recorded frequencies of the *Kdr* L995F (13–50%) remain lower than those previously reported in *An. gambiae* s.l. populations from urban and agro-industrial settings in southern Cameroon (up to 90% frequency). Although the *Kdr* L995S allele was not found in the samples, previous studies revealed that the latter mutation originally described in East Africa has already migrated to Central and West Africa [38,39,40].

The increase of P450 and COE activities and *Kdr* L995F frequencies recorded in this study can result from the wide use of insecticides in agriculture, public health, and domestic hygiene. The role of agricultural use of insecticides on the selection of *An. gambiae* s.l. resistance to insecticides is well documented [37,40]. Furthermore, the two nationwide LLINs distribution campaigns launched in Cameroon in 2011 and 2015, could have increased the selection pressure on local malaria vector populations. Before the LLIN distribution campaign in 2011, the coverage of vector control (ITNs/IRS) in rural areas of Cameroon was estimated at 10% [41]; net use was also very low in rural areas [42]. Indeed, the selection pressure originating from high LLIN coverage on top of pyrethroids used in urban agriculture and domestic hygiene in Graoua HD settings could explain the high resistance frequencies recorded in the Ouro-Housso/Kanadi urban area, compared with the Be-Centre sub-urban and Bala rural areas [28].

Selection pressure is a major force in evolution [43]. It can take many forms and can affect one, or several loci at the same time and act at different levels with different effects and interactions. Previous studies have anticipated a complex interplay between *Kdr*- and metabolic-based resistance mechanisms in *An. gambiae* s.l. from Cameroon [16]. The current study shows that deltamethrin resistance in semi-urban and rural areas of North Cameroon is now driven mainly by the *Kdr* L995F allele, in addition to P450 and COE detoxification. This is an example of the developmental process of multiple resistance mechanisms in the malaria vectors of the *An. gambiae* complex, which may also happen in other malaria vector species and in other African countries. The increasing frequencies of *Kdr* 995L/F heterozygotes and the subsequent increase in the *Kdr* L995F allelic frequencies in the presence of metabolic resistance enzymes (P450s and COEs) suggests a recent acquisition and possible overdominance of this resistance gene in the surveyed *An. gambiae*, *An. arabiensis* and *An. coluzzii* populations. This leads to the question of whether *Kdr* 995L/F heterozygous mosquitoes have higher fertility than their homozygous counterparts. Such heterozygote advantages could explain the increase of their frequencies from generation to generation despite ongoing selection pressure in the surveyed locations. Previous studies revealed that *Kdr* heterozygous males of *An. coluzzii* collected from mating swarms in Burkina Faso were more likely to successfully mate than homozygote resistant ones, illustrating a deleterious effect of homozygote resistant *Kdr* allele on *An. coluzzii* paternity success. Furthermore, these male mosquitoes were more competitive compared to homozygous-susceptible ones, indicating a heterozygous fitness advantage [44]. Other laboratory studies showed that the *Kdr* L995F allele positively affects *An. gambiae* life traits in terms of larval survivorship, pupation rate, and blood-feeding success, especially in heterozygote individuals [45]. Still, the fitness effects of *Kdr* L995F allele in natural populations of *An. gambiae* are to further be investigated.

On the operational side, early reduction in ITN effectiveness (i.e., after only 3 months of use) was previously associated with metabolic-based pyrethroid resistance in *An. gambiae* s.l. from North Cameroon [46]. The rapid increase in the frequency of *Kdr* L995F beside metabolic-based pyrethroid resistance can accelerate the decline of the effectiveness of LLINs in this region. Further studies are needed to understand the evolutionary trends of multiple insecticide resistance mechanisms in other regions of Cameroon and design the national strategic plan for resistance management in the coming years.

## 4. Materials and Methods

### 4.1. Study Period and Sites

The study was conducted during the rainy seasons, between September and October during two non-consecutive years (2011 and 2015), in 3 locations across three health districts (HD) of the North Cameroon Region (Figure 4), namely Ouro-Housso/Kanadi (9°18′ N; 13°24′ E) in the Garoua HD, Be-Centre (9°24′ N; 13°31′ E) in the Pitoa HD and Bala (9°46′ N; 13°44′ E) in the Mayo Oulo HD. Garoua is the capital city of the North Cameroon Region; Pitoa and Mayo Oulo are located 18 km in the North-East and 90 km in the North of Garoua, respectively. The 3 study sites are located alongside the Bénoué River basin. The climate in these areas is tropical Sudanese characterized by two seasons: one dry season extending from November to May and one rainy season from June to October, with an average annual rainfall between 700–1000 mm and a mean annual temperature of 28.1 °C [47].

The 3 study sites display different ecological features, including one urban area (Ouro-Housso/Kanadi), one peri-urban setting (Be-Centre), and one rural area (Bala). These sites have been classified according to the rates of built-up land (RBL), i.e., RBL > 50% for urban areas, RBL < 40% for rural areas; 40 < RBL < 50 for peri-urban areas proximate to the city. Crop growing activities are prominent in these HDs, with irrigated rice fields and intensive cotton farming in Pitoa, maize, and peanut cultivation in Mayo-Oulo; while corn, tomatoes, and eggplant are mainly grown in Garoua. The proportions of households owning at least one LLIN (mainly PermaNet 2.0 containing 1.4g/kg ± 25% deltamethrin) was 60–71%, and the LLINs utilization was 43–54% following the 2011 free LLIN distribution campaign [48]. Before the LLIN distribution campaign, the overall proportion of households owning at least one LLIN in the North Cameroon Region was estimated at 48%; however, net utilization was very low (≈13%) [41,42].

Malaria transmission in the study areas is seasonal with a yearly peak of transmission between September and October. Three species of the *An. gambiae* complex (*An. arabiensis*, *An. coluzzii*, *An. gambiae*) and *An. funestus* are responsible for most of the *Plasmodium* parasite transmission, with entomological inoculation rates (EIR) up to 1.2 infective bites/person/night (ib/p/n). *An. arabiensis* and *An. coluzzii* are the most infectious vector species in these areas. Other species, such as *An*. *pharoensis* and *An. rufipes* play secondary roles in the transmission [49]. *Plasmodium* infection rates in children under five years old were estimated at 48%, 26%, and 18%, in Pitoa, Garoua, and Mayo Oulo HD, respectively. *Plasmodium falciparum* accounts for most of the infections (91.6%), with *P. malariae* at a very low frequency (2.37%) [50].

### 4.2. Mosquito Collection and Processing

Mosquito larvae and pupae were collected from temporary and permanent water bodies (pools, footprints, flooded rice paddies, riverbeds, puddles in the corn and cotton fields) across the three study sites, using the dipping technique [51]. Collected immature mosquitoes were brought to local insectaries, fed with TetraMindBaby^®^ (fry food for larvae), and reared until adult emergence. Adults were fed with 10% glucose solution. They were identified using morphological identification reference keys [52,53], and only female *An. gambiae* s.l. were selected for use in deltamethrin susceptibility tests and molecular analyses.

### 4.3. Bioassays

With mosquito samples from each study site, three assays were conducted:One assay with sub-samples of mosquitoes not exposed to synergists prior to susceptibility tests, for establishing their resistance status to deltamethrin;Two assays with sub-samples of mosquitoes pre-exposed to synergists for one hour, i.e., piperonyl butoxide (PBO, Sigma Milwaukee, WI) and SSS-tributyl-phosphoro thrithioate (DEF, Sigma Milwaukee, WI), for detection of increased activity of two detoxifying enzymes, namely cytochrome P450 oxidases and carboxylesterases, respectively.

Filter papers impregnated with 0.05% deltamethrin were supplied by the Vector Control Research Unit of University Sains Malaysia (Penang, Malaysia). Synergist impregnated papers, i.e., 4% PBO and 0.25% DEF were prepared by the research team at the Malaria Research Laboratory of the Organization for the Coordination of Endemic Disease Control in Central Africa (OCEAC) (Yaoundé, Cameroon).

All bioassays (following exposure to synergists or not) were performed with unfed female mosquitoes aged 3–4 days, under ambient room temperature ranging (25–28 °C) and 70–80% relative humidity, using WHO susceptibility test kits and a standard protocol for adult mosquitoes [30]. Each complete set of bioassays was performed with five batches of 20–25 unfed females, four batches were exposed to insecticide-impregnated filter papers and one batch was exposed to untreated filter paper as a control.

During exposure to insecticides, the number of mosquitoes knocked down was recorded at 5 min intervals. After 1 h exposure, mosquitoes were transferred to holding tubes and provided with cotton pads soaked with 10% sugar solution. The mortality rates were determined 24 h post-exposure.

At each field survey, susceptibility tests were also conducted with 3–5-day-old unfed females of the Kisumu susceptible reference colony of *An. gambiae* maintained in the OCEAC Laboratory of Malaria Research, for more than 25 years.

Mosquitoes used as control batches during susceptibility tests, as well as dead and survivor mosquitoes 24 hours post contact with deltamethrin were kept separately at −20 °C for molecular identification of species and *Kdr* genotyping. Each mosquito was individually stored desiccated in labeled Eppendorf tubes.

## 5. Species Identification and *Kdr* Genotyping

Total DNA of mosquitoes used as control batches during susceptibility tests was extracted using the CTAB (cetyl trimethyl ammonium bromide) method [54]. Then mosquito species were identified using polymerase chain reaction-restriction fragment length polymorphism (PCR-RFLP) [55]. The *Kdr* L995F and L995S alleles were genotyped using hot oligonucleotide ligation assay (HOLA) as described by Lynd et al. [56].

## 6. Data Analysis

The knockdown times for 50% and 95% mosquitoes (Kdt_50_ and Kdt_95_) during exposure to deltamethrin were estimated using a log-probit model [57] and analyses were performed using the WIN DL (version 2.0, 1999) software. The rate of suppression of knockdown time by synergists (Kdts) was calculated according to the equation of Thomas et al. [58], which is: Kdts_50_ suppression (%) = [1 − (Kdt_50_ in presence of synergist/Kdt_50_ in absence of synergist)] × 100, with effective values above 10%.

The resistance status was defined according to the WHO criteria [30]; i.e., mortality rates less than 90% were indicative of resistance, mortality rates between 90% and 97% suggested suspected resistance to be confirmed, while mortality equal to or greater than 98% indicated susceptibility. Allelic frequencies at the *Kdr* 995 locus were calculated using Genepop Online (Version 4.5.1) [59]. Comparisons of mosquito mortality rates to deltamethrin with versus without synergists and the *Kdr* L995F/S allelic frequencies were performed using the Mantel–Haenszel chi-square test.

## 7. Conclusions

Data from this study show two evolutionary trends of *An. gambiae* s.l. resistance mechanisms to deltamethrin. In the Ouro-Housso/Kanadi urban area, there was a slow emergence of metabolic-based resistance while the frequency of the *Kdr* resistance L995F allele was regressing. Conversely in the Be-Centre peri-urban and Bala rural areas, there was a significant increase of the *Kdr* L995F frequencies, in parallel with the regression of metabolic-based resistance. Combined interventions including PBO bed nets or bed nets impregnated with new insecticides, such as chorphenapyr (class: pyrolle), IRS with clothianidin-containing products (class: neonicotinoid), or any other complementary interventions (larvicides) should be considered for the management of insecticide resistance in this region.

## Figures and Tables

**Figure 1 pathogens-11-00253-f001:**
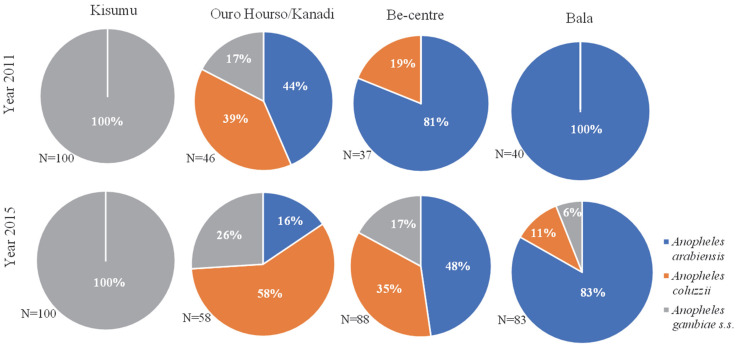
Species distribution among *Anopheles gambiae* s.l. populations collected in 2011 and 2015.

**Figure 2 pathogens-11-00253-f002:**
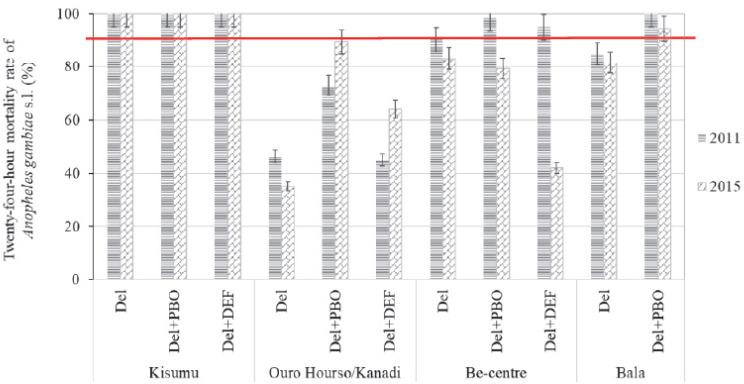
Twenty-four-hour mortality rates (with a confidence interval at 95%) of *Anopheles gambiae* s.l. following exposure to 0.05% deltamethrin with or without 4% Piperonyl butoxide (PBO) and 7% SSS-tributyl-phosphoro-thrithioate (DEF) in 2011 and 2015. The resistance status was defined according to the WHO criteria [30]; i.e., mortality rates less than 90% were indicative of resistance, mortality rates between 90% and 97% suggested suspected resistance to be confirmed, while mortality equal to or greater than 98% indicated susceptibility. The red line at 90% mortality rate indicates the threshold for confirmed insecticide resistance. Comparisons of mosquito mortality rates to deltamethrin with versus without synergists were performed using the Mantel–Haenszel chi-square test.

**Figure 3 pathogens-11-00253-f003:**
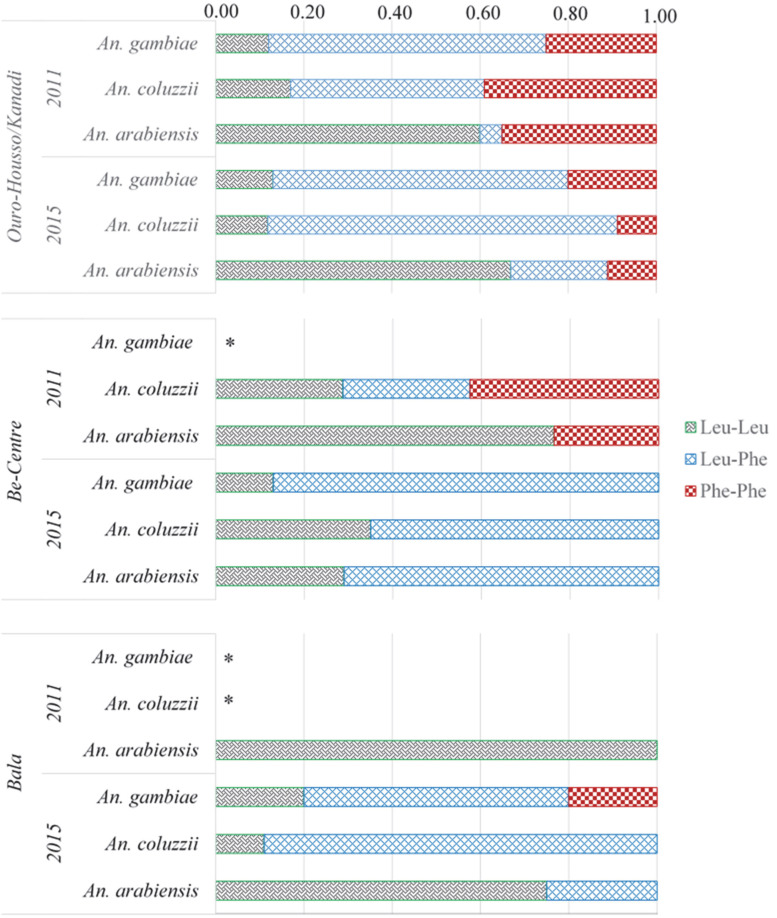
Frequency distribution of the different genotypes (per species) at position 995 of the *Kdr* locus among field-collected *An. gambiae* s.l. populations. Leu: Leucine position at position 995 (encoded by allele L995L); Phe: Phenylalanine position at position 995 (encoded by allele L995F); * Not determined because this species was not found.

**Figure 4 pathogens-11-00253-f004:**
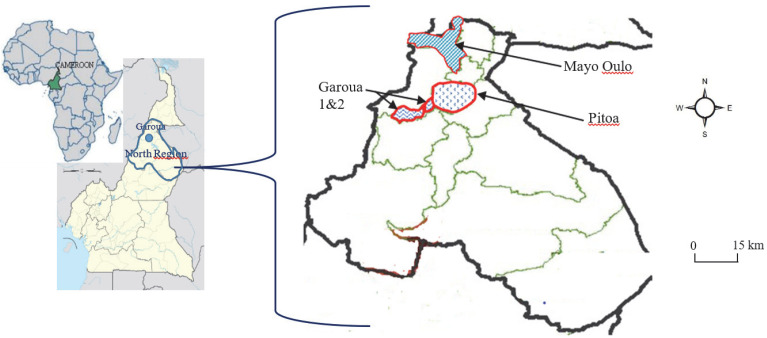
Map of the North Cameroon Region showing the study health districts.

**Table 1 pathogens-11-00253-t001:** Knockdown times and resistance status of *Anopheles gambiae* s.l. samples exposed to 0.05% deltamethrin.

Locality and District	Year	N	Tkd50 (CI), min	Tkd95 (CI), min	Resistance Status
Kisumu Reference strain	2011	86	9.5 (8.4–10.8)	17.3 (15.7–19.4)	S
	2015	86	9.4 (7.4–11.8)	18.6 (14.7–19.7)	S
Ouro-Housso/Kanadi Garoua district	2011	93	51.5 (30.7–64.2)	>60	R
	2015	100	>60	>60	R
Be-Centre Pitoa district	2011	82	23.5 (20.7–26.1)	>60	SR
	2015	95	37.4 (32.3–44.0)	>60	R
Bala Mayo-Oulo district	2011	75	41.2 (37.0–45.3)	>60	R
	2015	87	27.6 (25.9–29.3)	>60	R

Resistance status was defined according to the WHO criteria [30]; i.e., mortality rates less than 90% were indicative of resistance, mortality rates between 90% and 97% suggested suspected resistance to be confirmed, while mortality equal to or greater than 98% indicated susceptibility. N: sample size; Tkd50 and Tkd95: knockdown times for 50% and 95% of the tested population; CI: confidence interval at 95%; min: minute; R: resistant; S: susceptible; SR: suspected resistance.

**Table 2 pathogens-11-00253-t002:** Knockdown times and resistance status of *Anopheles*
*gambiae* s.l. samples to deltamethrin with and without synergists.

Locality and District	Year	Insecticide	N	Kdt_50_ (CI) (min)	Kdt_95_ (CI) (min)	Kdts_50_ (%)	Resistance Status
Kisumu Reference strain	2011	Del	86	9.5 (8.4–10.8)	17.3 (15.7–19.4)	-	S
		Del+PBO	88	9.2 (7.8–10.8)	18.4 (17.2–20.4)	3.2	S
		Del+DEF	82	8.8 (7.1–9.3)	16.8 (14.5–20.3)	7.4	S
	2015	Del	86	9.4 (7.4–11.8)	18.6 (14.7–19.7)	-	S
		Del+PBO	90	8.9 (6.8–10.3)	19.4 (16.2–21.3)	5.3	S
		Del+DEF	84	8.6 (7.1–10.7)	18.8 (15.5–20.6)	8.5	S
Ouro-Housso/Kanadi Garoua district	2011	Del	93	51.5 (30.7–64.2)	>60	-	R
		Del+PBO	81	28.7 (21.0–35.0)	>60	44.3	R
		Del+DEF	70	59.1 (47.3–24,854.6)	>60	<0	R
	2015	Del	100	>60	>60	-	R
		Del+PBO	68	40.3 (35.4–46.8)	>60	55.5	R
		Del+DEF	74	52.4 (45.2–68.7)	>60	30.0	R
Be-Centre Pitoa district	2011	Del	82	23.5 (20.7–26.1)	>60	-	SR
		Del+PBO	70	10.8 (6.6–14.6)	41.8 (28.4–94.4)	54.0	S
		Del+DEF	86	6.8 (4.3–8.8)	24.7 (18.6–42.3)	71.1	SR
	2015	Del	95	37.4 (32.3–44.0)	>60	-	R
		Del+PBO	82	14.9 (12.0–17.7)	>60	60.2	R
		Del+DEF	88	>60	>60	<0	R
Bala Mayo Oulo district	2011	Del	75	41.2 (37.0–45.3)	>60	-	R
		Del+PBO	51	38.1 (26.1–101.2)	50 (23.0–249.0)	7.5	S
	2015	Del	87	27.6 (25.9–29.3)	>60	-	R
		Del+PBO	89	26.3 (24.6–28.0)	>60	4.7	SR

N: sample size; Del: deltamethrin; PBO: piperonyl butoxide; Kdt_50_ and Kdt_95_: knockdown times for 50% and 95% of the tested population; Kdts_50_: percent of knockdown time suppression; CI: confidence interval at 95%; min: minute; Tkd50R: time ratio required for knocking down 50% of individuals; R: resistant; S: susceptible; SR: suspected resistance.

**Table 3 pathogens-11-00253-t003:** Allelic frequencies (per species) at the *Kdr* 995 locus in field *Anopheles gambiae* s.l. samples in 2011 and 2015.

Locality and District	Year	Species	N	Fs L995F (%)	Fc L995F (%)	*p* Value
Ouro-Housso/Kanadi Garoua district	2011	*An. arabiensis*	20	37.50a	50.00	0.88
		*An. coluzzii*	18	61.11c		
		*An. gambiae*	8	56.25g		
	2015	*An. arabiensis*	9	22.22a	45.69	
		*An. coluzzii*	34	48.53c		
		*An. gambiae*	15	53.33g		
Be-Centre Pitoa district	2011	*An. arabiensis*	30	3.33a	13.5	0.02
		*An. coluzzii*	7	57.14c		
		*An. gambiae*	0	-		
	2015	*An. arabiensis*	42	35.71a*	35.80	
		*An. coluzzii*	31	32.26c		
		*An. gambiae*	15	43.33		
Bala Mayo-Oulo district	2011	*An. arabiensis*	40	0.00a	0.00	0.01
		*An. coluzzii*	0	-		
		*An. gambiae*	0	-		
	2015	*An. arabiensis*	69	12.32a*	18.10	
		*An. coluzzii*	9	44.44		
		*An. gambiae*	5	50.00		

N: sample size; SS: homozygote susceptible; SR: heterozygote; RR: homozygote resistant; Fs: allelic frequencies of *Kdr* 995F of each species *An. arabiensis*, *An. coluzzii* and *An. gambiae* (*s.s*.); Fc: overall allelic frequencies of the *Kdr* L995F for the three identified species of the *An. gambiae* complex; a: non-significant difference in the *Kdr* L995F allelic frequency in *An*. *arabiensis;* c: non-significant difference in the *Kdr* L995F allelic frequency in *An*. *coluzzii;* g: non-significant difference in the *Kdr* L995F allelic frequency in *An*. *gambiae;* a*: significant difference in the *Kdr* L995F allelic frequency in *An*. *arabiensis.*

**Table 4 pathogens-11-00253-t004:** Putative insecticide resistance mechanisms in wild *An. gambiae* s.l. samples in 2011 and 2015.

Locality	Year	Species and *Kdr* L995F freq.	PBO Effect		DEF Effect		*Kdr* L995F Allele	Resistance Status
			Kdts_50_	Mrv.	Kdts_50_	Mrv.		
Ouro-Housso/Kanadi	2011	Aa **, Ac **, Ag **	+	+	-	-	+++	R
	2015	Aa *, Ac **, Ag **	++	++	+	+	++	R
Be-Centre	2011	Aa *, Ac ***	++	+	++	+/-	++	RP
	2015	Aa **, Ac **, Ag **	++	-	-	-	+++	R
Bala	2011	Aa	+/-	+	N/A	N/A	-	R
	2015	Aa *, Ac **, Ag **	+/-	+	N/A	N/A	+++	R

Aa: *Anopheles arabiensis*, Ac: *Anopheles coluzzii*; Ag: *Anopheles gambiae*; kdts_50_: knockdown time suppression; Mrv: mortality reversion; * presence of *Kdr* L995F at <25% frequency; ** presence of *Kdr* L995F at 25–40% frequency; *** presence of *Kdr* L995F at >50% frequency; +freq: Frequency; -: not detected; +/-: possible role in resistance to be confirmed; +: minor role in resistance; ++: Moderate role in resistance; +++: Major role in resistance; N/A: not applicable; R: confirmed resistance; SR: suspected resistance.

## Data Availability

All data generated or analyzed during the current study are included in this published article.

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
