# Peer review of "Patterns of *Kdr*-L995F Allele Emergence Alongside Detoxifying Enzymes Associated with Deltamethrin Resistance in *Anopheles gambiae* s.l. from North Cameroon"

_pathogens, 2022, doi:10.3390/pathogens11020253_

Round 1
Reviewer 1 Report
Authors of this article collected adequate number of mosquitoes to be tested for their susceptibility to deltamethrin (as indicated in tables 1 and 2) in each of the site. It is not clear as to why they used very few mosquitoes to show the distribution of Anopheles gambiae s.l. in these sites (figure 2). The sample size used cannot give the true picture of mosquito species distribution across the study sites. This should be stated somewhere in the manuscript. Otherwise, the manuscript is well written.
Author Response
We are grateful to the reviewer for his valuable comments which help us to improve the quality of the manuscript. The issue raised has been addressed.
Reviewer 1
Comment 1: Authors of this article collected adequate number of mosquitoes to be tested for their susceptibility to deltamethrin (as indicated in tables 1 and 2) in each of the site. It is not clear as to why they used very few mosquitoes to show the distribution of Anopheles gambiae s.l. in these sites (figure 2). The sample size used cannot give the true picture of mosquito species distribution across the study sites. This should be stated somewhere in the manuscript. Otherwise, the manuscript is well written.
Response 1: We thank the reviewer for this comment. A total of 352 field mosquitoes were morphologically identified to species and the sample size per study period and site, generally varied between 37 and 88. These mosquitoes were those used as control during susceptibility test, in order to reflect the composition of samples that were exposed to deltamethrin. And there was no mosquito left for increasing the sample size, considering the number of tests that had to be performed at each survey and the low density of larvae in breeding sites. Nevertheless, the distribution of species was similar to that reported in previous studies. This issue has been stated in the discussion section, page 11, lines 371-390.

Reviewer 2 Report
Review of: Etiang et al. 2022 Pathogens
Evolutionary Trends of Kdr-L995F Allele and Detoxifying Enzymes Shaping Deltamethrin Resistance in Anopheles gambiae s.l. from North-Cameroon.
The manuscript describes a detailed vector and insecticide resistance survey in North Cameroon. The authors used larval sampling for collections. Adult specimens were identified by morphological and genetic means, and bioassays were carried out to test for pyrethroid resistance. The underlying mechanisms of resistance were established by use of synergists for cytochrome P450s and carboxylesterases, and the testing of kdr mutation frequency. These studies remain relevant and useful at the national level so that vector control measures can be appropriately implemented or modified as required. That said, the authors attempt to establish an evolutionary trend based on the data collected. I don’t believe that the findings support this and no cause and effect/correlation relationships were established (full resistance was present in two of the three populations when the studied commenced, and partial resistance in one). In this light, the title should be changed accordingly.
- Some specific comments:
There are grammatical errors throughout, and these must be corrected. For example:
Line 57-58: “…an acute febrile illness caused by…”
Line 59: “…use of pyrethroid insecticides on…”
Line 70: “…are strongly selected for by…”
Line 70-71: Resistance has absolutely reduced the efficacy of IRS and LLIN’s so I would reword this slightly.
Line 91: Delete “both”
Line 271 – decrease in genotypic frequencies from 25 to 40% in 2011 and 9 to 20% in 2015
Line 325: “…exhibiting multiple insecticide resistance mechanisms”.
Line 392: “…(genomic, individual and population)”. (Delete “…”)
Line 453: Delete “etc.”, and complete fully.
- Other Concerns:
Lines 1-3: Title: I think that the data is too simplistic to show real processes of evolution and what is causing them in these populations. Full resistance existed in two sites already. Species distribution changed somewhat, but the mechanisms underlying the resistant phenotype remained largely unchanged. The title must be amended to accommodate this.
Figure 2 - Add % values to the relevant sections of each pie chart
Figure 3 – Legend and axis: check abbreviation DEF (it doesn’t seem correct for SSS tributyl phosphoro 464 thrithioate).
Figure 3 – No red line; legend and y-axis – please include that this represents 24-hour mortality. The statistical analysis conducted for these data must be included on the figure.
Figure 3 - How is the big decrease in mortality of Be-Centre mosquitoes with Del + DEF explained? (it is much lower than Deltamethrin only exposure) – address this. This seems more like experimental error (if the carboxylesterases have no impact on resistance status you would expect mortality to be the same as that of the control).
Line 200 – I don’t think synergists will assist with early detection. As you demonstrate, resistance was already detected in 2011.
Table 4 - I find the presentation a little tricky to decipher. I feel this can be improved and the results better explained in the text.
Line 390: Selection pressure is a major force in evolution. (Delete “obviously” and include citations for this statement).
Lines 402-405: This is inaccurate… see Medigbododo et al., 2021; other citations on the topic are available too and should be added.
Author Response
We are grateful to the reviewer for his valuable comments which help us to improve the quality of the manuscript. Most of the issues raised have been addressed; for those that we have not fully considered, we provided some explanations.
Reviewer 2
Evolutionary Trends of Kdr-L995F Allele and Detoxifying Enzymes Shaping Deltamethrin Resistance in Anopheles gambiae s.l. from North-Cameroon.
The manuscript describes a detailed vector and insecticide resistance survey in North Cameroon. The authors used larval sampling for collections. Adult specimens were identified by morphological and genetic means, and bioassays were carried out to test for pyrethroid resistance. The underlying mechanisms of resistance were established by use of synergists for cytochrome P450s and carboxylesterases, and the testing of kdr mutation frequency. These studies remain relevant and useful at the national level so that vector control measures can be appropriately implemented or modified as required.
Comment 1: That said, the authors attempt to establish an evolutionary trend based on the data collected. I don’t believe that the findings support this and no cause and effect/correlation relationships were established (full resistance was present in two of the three populations when the studied commenced, and partial resistance in one). In this light, the title should be changed accordingly.
Response 1: The current study is actually the first attempt to establish the evolutionary trends of two major groups of pyrethroid resistance mechanisms in Anopheles gambiae from North Cameroon, thus highlighting the originality of this study. The three study sites were selected based on their resistance profiles in 2011. According to the WHO susceptibility testing protocol, a mortality rate less than 90% is suggestive of confirmed resistance. However, considering the large interval of mortality between 0 and 90%, a mosquito population classified as resistant may be subject to a gradual decline of mortality rates associated with an increase in resistance intensity, to be confirmed using 5X or 10X diagnostic concentrations. For instance in Houro-Ousso where the mosquito mortality rate was about 50% in 2011, there was a decrease of mortality in 2015. The objective of this study was to assess the changes in the two major types of resistance mechanisms over time, knowing that these mechanisms actually shape the phenotype of deltamethrin resistance. We observed variations in three tested mosquito populations, including the samples from the Be-centre that displayed probable resistance at the baseline survey in 2011. Other approaches can also be used to highlight the evolutionary trends of resistance in field mosquito populations, but we believe that the conclusions of the present study are in line with the obtained results. However, to address the reviewer's comment, we have proposed an alternative title: “Patterns of Kdr-995F Allele emergence alongside Detoxifying Enzymes associated with Deltamethrin Resistance in Anopheles gambiae s.l. from North-Cameroon”.
- Some specific comments:
There are grammatical errors throughout, and these must be corrected. For example:
Comment 2: Line 57-58: “…an acute febrile illness caused by…”
Response 2: The text has been revised accordingly, line 58.
Comment 3: Line 59: “…use of pyrethroid insecticides on…”
Response 3: The text has been revised accordingly, line 59.
Comment 4: Line 70: “…are strongly selected for by…”
Response 4: The text has been revised accordingly, lines 70-71.
Comment 5: Line 70-71: Resistance has absolutely reduced the efficacy of IRS and LLIN’s so I would reword this slightly.
Response 5: The text has been revised accordingly, lines 71-72.
Comment 6: Line 91: Delete “both”
Response 6: The text has been revised accordingly, line 91.
Comment 7: Line 271 – decrease in genotypic frequencies from 25 to 40% in 2011 and 9 to 20% in 2015
Response 7: There was a mix-up in here, we have now rephrased, lines 294-295.
Comment 8: Line 325: “…exhibiting multiple insecticide resistance mechanisms”.
Response 8: The text has been revised accordingly, line 357.
Comment 9: Line 392: “…(genomic, individual and population)”. (Delete “…”)
Response 9: The text has been revised accordingly, line 438.
Comment 10: Line 453: Delete “etc.”, and complete fully.
Response 10: The text has been revised accordingly, line 510.
- Other Concerns:
Comment 11: Lines 1-3: Title: I think that the data is too simplistic to show real processes of evolution and what is causing them in these populations. Full resistance existed in two sites already. Species distribution changed somewhat, but the mechanisms underlying the resistant phenotype remained largely unchanged. The title must be amended to accommodate this.
Response 11: See response to comment , lines 1-3.
Comment 12: Figure 2 - Add % values to the relevant sections of each pie chart.
Response 12: The percentages have been added on Figure 2.
Comment 13: Figure 3 – Legend and axis: check abbreviation DEF (it doesn’t seem correct for SSS tributyl phosphoro 464 thrithioate).
Response 13: The text has been revised accordingly.
Comment 14: Figure 3 – No red line; legend and y-axis – please include that this represents 24-hour mortality. The statistical analysis conducted for these data must be included on the figure.
Response 14: Figure 3 footnote and y-axis have been revised accordingly.
Comment 15: Figure 3 - How is the big decrease in mortality of Be-Centre mosquitoes with Del + DEF explained? (it is much lower than Deltamethrin only exposure) – address this. This seems more like experimental error (if the carboxylesterases have no impact on resistance status you would expect mortality to be the same as that of the control).
Response 15: In essence, we would expect the deltamethrin-DEF mortality rates to be comparable to that of deltamethrin without DEF, in case carboxylesterases have no impact on resistance status. However, this was not the case; DEF rather acted as a deltamethrin resistance stimulus. Though this data could result from an experimental error, similar results have been recorded in previous studies (Nwane et al. 2013). Further investigations are needed to better understand the interactions between insecticides and synergists. This statement has been included in the discussion (lines 409-416).
Comment 16: Line 200 – I don’t think synergists will assist with early detection. As you demonstrate, resistance was already detected in 2011.
Response 16: The word “early” has been removed from the text, line 223.
Comment 17: Table 4 - I find the presentation a little tricky to decipher. I feel this can be improved and the results better explained in the text.
Response 17: We wanted to show the kdr 995F frequencies in each of the three species of the An. gambiae complex and then in the overall samples of the complex used for susceptibility tests. This is because morphological identification of mosquito used for susceptibility tests only allows to identify of species complex and not individual species. So, the resulting phenotype of resistance is a combined outcome of the three species. We have provided clarifications in the text, in order to make the table more understandable.
Comment 18: Line 390: Selection pressure is a major force in evolution. (Delete “obviously” and include citations for this statement).
Response 18: The word “obviously” has been deleted, and a reference added, line 337.
Comment 19: Lines 402-405: This is inaccurate… see Medigbodo et al., 2021; other citations on the topic are available too and should be added.
Response 19: The text has been revised accordingly and 2 references added, line 450-461.

Reviewer 3 Report
Dear authors,
This paper deals with the patterns of the kdr995F resistance allele emergence alongside metabolic-based resistance to pyrethroids in North Cameroon, following the first large-scale distribution of LLINs.
Further down, you will find some comments of minor and major importance that may benefit the manuscript.
Lines 123-136: This paragraph should be modified in order to describe the aim of the current work, i.e. the distribution of resistance mechanisms in Anopheles gambiae s.l. mosquitoes in North Cameroon during 2011 and 2015, not the findings. The text in lines 327-330 could be moved here as more relevant for the scope of the manuscript.
Line 440: change “law” to “low”.
Fig. 3: Please indicate clearly in the figure the red line for 90% mortality threshold.
Table 2: I propose to add in the table % mortality values for each tested sample indicating significant differences since they subjected to statistical analysis and they are mentioned in the results section of the manuscript.
Table 3: The allelic frequencies for each species in each year could be subjected to statistical analysis (e.g. x2 test) to compare the frequencies between the years, as mentioned in the text of the results. Significant differences should be indicated in the table.
Lines 306-308: This sentence could be rephrased since the increase of Kd and mortality normally suggest the decrease of metabolic detoxification, not the upsurge as stated.
Table 4: In the footnote add (+) and the explanatory meaning
Discussion
The results of this study (lines 190-193) suggest high resistance frequencies in urban areas compared to sub-urban and rural areas. Could you please justify this finding and discuss it in few lines using relevant literature data, if available? Is it because of more intense use of pyrethroids through the use of LLINs and IRS in urban areas? Is indeed the use of pyrethroids in urban areas more intense compared to the agricultural use?
Lines 403-405: A possible explanation related to fertility is provided for the increase of heterozygous resistant genotypes and decrease of homozygous resistant genotypes over time. Could you please support this explanation with some relevant literature data?
Author Response
We are grateful to the reviewer for his valuable comments which help us to improve the quality of the manuscript. Most of the issues raised have been addressed; for those that we have not fully considered, we provided some explanations.
Reviewer 3
This paper deals with the patterns of the kdr995F resistance allele emergence alongside metabolic-based resistance to pyrethroids in North Cameroon, following the first large-scale distribution of LLINs.
Further down, you will find some comments of minor and major importance that may benefit the manuscript.
Comment 1: Lines 123-136: This paragraph should be modified in order to describe the aim of the current work, i.e. the distribution of resistance mechanisms in Anopheles gambiae s.l. mosquitoes in North Cameroon during 2011 and 2015, not the findings. The text in lines 327-330 could be moved here as more relevant for the scope of the manuscript.
Response 1: The text has been revised accordingly, lines 123-125.
Comment 2: Line 440: change “law” to “low”.
Response 2: The text has been revised accordingly, line 496.
Comment 3: Fig. 3: Please indicate clearly in the figure the red line for 90% mortality threshold.
Response 3: The red line for 90% mortality threshold has now been added to Fig. 3.
Comment 4: Table 2: I propose to add in the table % mortality values for each tested sample indicating significant differences since they subjected to statistical analysis and they are mentioned in the results section of the manuscript.
Response 4: Since the mortality rates are presented in Figure 3, with confidence intervals, we thought that adding them again in the table would be a repetition. That's why we have completed Table 2 with the statues of the resistance instead.
Comment 5: Table 3: The allelic frequencies for each species in each year could be subjected to statistical analysis (e.g. x2 test) to compare the frequencies between the years, as mentioned in the text of the results. Significant differences should be indicated in the table.
Response 5: Statistical analyses have been done; Table 3 has been revised accordingly and the text as well.
Comment 6: Lines 306-308: This sentence could be rephrased since the increase of Kd and mortality normally suggest the decrease of metabolic detoxification, not the upsurge as stated.
Response 6: This is an increase of knock-down suppression and mortality rates to deltamethrin, following pre-exposure to PBO and DEF. The text has been revised accordingly, line 339.
Comment 7: Table 4: In the footnote add (+) and the explanatory meaning
Response 7: Done
Discussion
Comment 8: The results of this study (lines 190-193) suggest high resistance frequencies in urban areas compared to sub-urban and rural areas. Could you please justify this finding and discuss it in few lines using relevant literature data, if available? Is it because of more intense use of pyrethroids through the use of LLINs and IRS in urban areas? Is indeed the use of pyrethroids in urban areas more intense compared to the agricultural use?
Response 8: These findings have now been discussed, lines 432-436.
Comment 9: Lines 403-405: A possible explanation related to fertility is provided for the increase of heterozygous resistant genotypes and decrease of homozygous resistant genotypes over time. Could you please support this explanation with some relevant literature data?
Response 9: Literature data have been provided, lines 450-461.

Round 2
Reviewer 3 Report
Dear authors,
Thank you for considering my comments. Your replies are sufficient and the manuscript is revised accordingly, so I propose its acceptance for publication.